# Interpretable Machine Learning with Brain Image and Survival Data

**Matthias Eder** [1], **Emanuel Moser** [2], **Andreas Holzinger** [2,3,4], **Claire Jean-Quartier** [3,5]
**and Fleur Jeanquartier** [3,*]

1 Institute of Software Technology, Graz University of Technology, 8010 Graz, Austria
2 Institute of Interactive Systems and Data Science, Graz University of Technology, 8010 Graz, Austria
3 Human-Centered AI Lab (Holzinger Group), Medical University of Graz, 8036 Graz, Austria
4 Institute of Forest Engineering, University of Natural Resources and Life Sciences, 1190 Vienna, Austria
5 Research Data Management, Graz University of Technology, 8010 Graz, Austria
* Correspondence: f.jeanquartier@hci-kdd.org

**Abstract:** Recent developments in research on artificial intelligence (AI) in medicine deal with the analysis of image data such as Magnetic Resonance Imaging (MRI) scans to support the of decision-making of medical personnel. For this purpose, machine learning (ML) algorithms are often used, which do not explain the internal decision-making process at all. Thus, it is often difficult to validate or interpret the results of the applied AI methods. This manuscript aims to overcome this problem by using methods of explainable AI (XAI) to interpret the decision-making of an ML algorithm in the use case of predicting the survival rate of patients with brain tumors based on MRI scans. Therefore, we explore the analysis of brain images together with survival data to predict survival in gliomas with a focus on improving the interpretability of the results. Using the Brain Tumor Segmentation dataset BraTS 2020, we used a well-validated dataset for evaluation and relied on a convolutional neural network structure to improve the explainability of important features by adding Shapley overlays. The trained network models were used to evaluate SHapley Additive exPlanations (SHAP) directly and were not optimized for accuracy. The resulting overfitting of some network structures is therefore seen as a use case of the presented interpretation method. It is shown that the network structure can be validated by experts using visualizations, thus making the decision-making of the method interpretable. Our study highlights the feasibility of combining explainers with 3D voxels and also the fact that the interpretation of prediction results significantly supports the evaluation of results. The implementation in python is available on gitlab as "XAIforBrainImgSurv".

**Keywords:** radiomics; survival prediction; glioma; interpretability; deep learning; convolutional neural networks; explainable artificial intelligence





## 1. Introduction

Radiomics is a non-invasive method supporting brain cancer diagnosis [1]. The discipline also stimulates growing image datasets as well as clinical studies involving such data for developing predictive models [2–5].

Cancers from the central nervous system consist of a heterogeneous group of tumors with very different biologies and prognoses [6]. Glioma presumably originates from glial cells that are neural stem cells, astrocytes derived therefrom, or oligodendrocyte precursor cells [7]. Brain cancers are regarded as the primary cause of morbidity and mortality, while glioma comprises the majority therefrom [8]. Glioma management is based on an integrated approach of clinical examination, brain imaging, and molecular feature analysis set by the international standard for the classification of brain and spinal cord tumors [9].

Glioma research still lacks data describing specific features in subtypes or clinical implications such as age groups [10]. Still, endeavors such as the RSNA-ASNR-MICCAI Brain Tumor Segmentation (BraTS) challenge initiative deliver benchmarks and also foster

interdisciplinary exchange. However, survival prediction of glioma patients remains a challenging task [11,12]. Several studies and reviews show the use of medical imaging techniques in combination with deep learning for various medical applications [13–17], such as tumor/tissue segmentation [18,19], anatomical/cell structure detection [20] or computed-aided diagnosis and prognosis [21]. Deep learning techniques and medical image analysis can also help to predict the survival rate of glioma patients [22]. However, deep learning approaches are typically hard to understand, lack interpretability and are therefore often seen as a black-box [23]. The field of explainable artificial intelligence (XAI) aims to open the black box of deep learning and to understand its internal decisions better [24–26] and is reported to be an important future direction for glioma survival analysis [27].

This study aims to explore possibilities to explain glioma survival prediction. While making use of open data convolutional neural networks (CNN) for training, complete 3D MRI scans are combined with an explanation approach that is compatible with 3D voxels. By including an additional pre-processing step to increase the size of the available dataset, it is aimed to improve the performance of the trained network models. The increasing complexity of the network model becomes apparent using complete 3D voxels as training data, and multiple models are examined by means of performance. To examine the applicability and validity of the trained network models, visual explanations are created using SHapley Additive exPlanations (SHAP) [28]. With these explanations, we aim to explore and identify influencing factors of the predicted survival rate based on the networks' inputs and increase the interpretability of the results. Through a comparison of the identified influencing factors with extracted domain knowledge, the used network can be assessed and validated without having to rely on additional validation datasets.

The main contributions of this work are (1) the investigation of the usage of full 3D MRI voxels to predict the survival rate of glioma patients and (2) the usage of XAI visualization techniques for evaluation of the trained network model and increasing its interpretability. Additionally, a proposal for a pre-processing step to augment 3D MRI scans and increase the number of data samples for training is presented.

The remainder of this work is structured as follows. First, related work in the field of survival prediction, deep learning on medical image analysis, and explainable artificial intelligence is given. Then, the concept for pre-processing and augmenting 3D MRI scans, as well as the used CNN structure for training, is presented. In Section 3, training results and extracted XAI features for enhancing interpretability are presented, which are discussed in Section 4. Section 5 concludes the paper.

### 1.1. Background on Methodical Approaches to Survival Prediction

Gliomas are diagnosed by several steps of medical history, physical and neurological examination, histopathological analysis for molecular features and radiographic examination, including magnetic resonance imaging (MRI) [29]. Grading and classification are based on histological features, which have been updated recently [6,30]. Conventional imaging analysis involves features such as the degree and heterogeneity of contrasted enhancement area of tumors, edema in surrounding tissues, hemorrhage, border definition, mass effect, or varying intensities in T2-weighted MRI, among others, as indicators for malignancy, while three-dimensional image analysis is recommended due to anisotropic growth of glioma [31–33].

Imaging after surgery is recommended to evaluate the extent of the resection; moreover, pathological enhancement thickness on post-surgical images could be correlated with survival [34].

Survival prediction in glioma is an ongoing research topic, often making use of radiomic and/or genomic and clinical data and applying ML methods [27]. Experiments with mostly (but not exclusively) machine learning (ML) approaches are conducted to see which one works best [35,36]. The BraTS 2018 dataset containing survival information has been used to predict the survival days via a linear model and comparing it to a neural network

and random forest [37]. The model was not directly trained on MRI scans but on features that were extracted from them. Similar to that, the survival rate has been predicted through an artificial neural network based on extracted features [38]. Although it is not specifically mentioned how features are extracted (manually vs. automatically), the authors showed that the used features were not applicable for survival prediction. An automatic extraction approach of MRI features of glioblastoma patients to predict the survival rate has been presented by [39]. Eight extracted image features were used in a Cox regression model and evaluated for survival correlation. While the presented approach shows how features can be extracted for survival prediction, it lacks a clear methodology, and no correlation results could be shown. Additional specific examples of survival-associated features extracted from MRI scans of glioma are given by [18,40], who identified subsets of 5 or 14 significant features, respectively, correlating to patients' survival by using data-mining techniques in combination with a J48 classification tree for prediction or decision trees for feature extraction, respectively, in combination with a random forest model for survival prediction.

Convolutional neural networks (CNNs) can be further used to extract deep MRI features for diagnosis in terms of classification [41].

### 1.2. Background on MRI Regression/Classification on CNNs

The classification and regression of glioma types next to the survival prediction with CNNs is another ongoing topic of radiomics on glioma MRI imaging. In [18], the authors use the BraTS 2018 dataset to conduct tumor segmentation on MRI scans using an ensemble of three different 3D CNN architectures. By combining all three networks, they managed to segment the tumor well and achieved a good performance on tumor segmentation. The authors in [22] classified survival predictions based on histopathological images using three classes of survival. They compared five different classifiers based on DCNNs and tested different patch sizes. They concluded a patch size of 256 to be optimal, offering a good training accuracy and a good loss curve. The work in [42] explores different pre-trained CNNs and classifiers to detect abnormalities in MRI brain images. In another experiment, the authors of [43] applied a CNN to classify the tumor type, distinguishing between glioma, meningioma, and pituitary using data augmentation techniques. As the input, 2D slices for MRI scans were collected and manually labeled. To increase the training size, the data set was additionally augmented by transforming and rotating the images before splitting. Although they did not evaluate how the inflation of the training set affects the results of their experiments, it can be assumed that the increase in the training set influences the accuracy. Another application of CNNs is the classification of IDH genotypes, as the distinction between IDH-wildtype and IDH-mutant, from dynamic susceptibility contrast (DSC) MRI images [44]. Therefore, they recorded and pre-processed MRI images by applying skull stripping, co-registration, bias field correction, and isotropic resampling. They then segmented the subregion of the tumor using neural networks and classified the two IDH genotypes using a convolutional long short-term memory (LSTM) network. Their results showed a high training accuracy. Here again, no evaluation of the influence of image pre-processing is conducted, but other research suggests that this approach enhances the training results significantly [45–47].

Other specific tumor classification approaches currently proposed are hybrid models, as CNN and neural autoregressive distribution estimation (NADE) [48], "AdaptAhead" Optimization for MRI segmentation [49] and 3D CNNs for brain tumor segmentation [50] or for tumor classification [51]. An extensive overview of the applications of deep learning on MRI data, illustrating a broad field of applications that is not limited to the brain and using different MRI images of the human body, is given in [52]. Their set of applications includes image registration, image segmentation, resolution improvement, quantitative parameter description, diagnosis, and prediction.

The different image modalities of fluid-attenuated inversion recovery (FLAIR), T1-, T2-weighted, and T1 contrast-enhancing (T1ce) in MRI are based on the chosen institutional methods and measurement parameters [53]. T1, T2 and FLAIR images result from either

short, long, or even longer times to echo (TE) and repetition times (TR) given through the application and receipt of varying sequences of radio frequency pulses [54]. Contrast enhancement agents further change signaling intensities by shortening T1 relaxation rates [55]. These modalities highlight the contrast between various tissue types differently. Tumor mass is less bright in T2 and FLAIR, while inverse for T1ce. Edema are bright in FLAIR. Necrosis is dark in T1 and FLAIR, and it is bright in T2 [53].

### 1.3. Explainable AI in MRI Imaging

Explainability, as defined by the XAI community, highlights technically decision-relevant parts of machine representations and machine models, for example, those parts that contributed to model accuracy in training or to a particular prediction. Importantly in our context, this definition does not refer to a human model. For this purpose, causability was introduced [56], following the notion of usability [57]. While XAI is about implementing transparency and traceability, causability is about measuring the quality of explanations [58], i.e., the measurable extent to which an explanation of a statement achieves a certain level of causal understanding for a user with effectiveness, efficiency, and satisfaction in a given context of use. According to the DIN EN ISO 9241-11 norm [59] describing the ergonomics in a human–system interaction, usability represents the measurable extent to which a software can be used by certain users to achieve certain goals with effectiveness, efficiency and satisfaction in a certain context of use, and causability is the measurable extent to which an explanation achieves a certain level of causal understanding for a human. Thus, it relates to a human model and attempts to make the causal relationships understandable to the domain expert in the sense of [60]. Different general methods to approach explainable AI models are presented and discussed in a use case on histopathology [24] and neuroimaging [61].

Currently, there is still a lack of explainability in models based on radiomics [62]. Moreover, explainability supports the evaluation process by making models understandable also to medical scientists, which is ultimately necessary for clinical validation and essential for AI as the decision support in regard to transparency of machine learning models [63]; an example is given by [64]. In [65], the authors show an application of CNN for image classification on breast MRI images with the addition of XAI. They adapted the AlexNet structure [66] for binary classification and applied an integrated gradients attribution method and SmoothGrad noise reduction for visualization of relevant features, opening the "black-box" results. Similar to that, class activation mapping (CAM) is used not only to obtain the prediction and the certainty score of a neural network but also to obtain the information on what regions of the input the result comes from by overlaying the input with a heatmap, indicating the regions of interest [67]. An XAI approach called Grad-CAM, which extends CAM by including gradient information, is presented in [68]. Another approach that is available under the open-source license is SHapley Additive ex-Planations (SHAP) [28]. The framework provides additive feature attribution methods and visualization of these, such as importance. The field of XAI has shown enormous potential, and new techniques, such as Grad-CAM or SHAP, are released every year. Other currently used techniques are *LIME* [69], *DeepLIFT* [70] and *CXplain* [71]. A different explanation extraction framework that answers a set of specific and humanistic questions verbally instead of visualizing the trained network is presented in [72].

Current state-of-the-art XAI python frameworks have been compared using a genomic classification of glioma subtypes [73], and medical imaging analysis and deep learning approaches in various fields have been summarized and surveyed [23,74].

## 2. Concept and Implementation

This section describes the process of how to create an interpretable machine learning (ML) framework that provides additional information about the trained network. Figure 1 abstracts the architecture graphically. First, the used dataset and the pre-processing steps are described. Then, the used ML approach, a CNN using voxels, is detailed. At the

end of this section, the interpretation method is presented. Implementation sources for pre-processing, training and validation are available via https://gitlab.com/matte3000/xai-for-brain-img-surv (created 29 April 2021, last updated 31 August 2022).

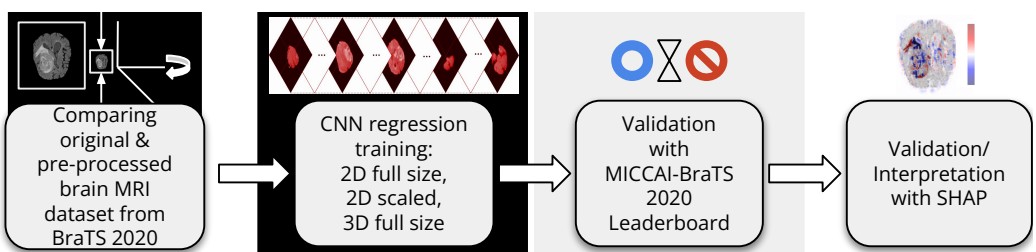

**Figure 1.** The architectural overview.

### 2.1. Data Pre-Processing

For training our ML approach, the BraTS2020 dataset is used [2,75,76], which contains survival information of high-grade glioma (HGG) and low-grade glioma (LGG) patients and comprises four image modalities of 3D MRI scans (t1, t1ce, t2, flair). We use those data from the original training dataset that include survival information in days. The provided data are already pre-processed by co-registering the data to the same anatomical template, interpolating them to the same resolution of 1 mm$^3$ and by conducting skull-stripping.

We further augment the selected data of 235 patients from the BraTS2020 dataset by implementing random rotation. This step is used to increase the size of the available dataset and improve training results. The MRI scans of the patients were randomly rotated around the longitudinal axis (roll $r$), transverse axis (pitch $p$), and vertical axis (yaw $y$) within the range of $-20° \leq r, p, y \leq 20°$. This was performed 10 times for every MRI scan, resulting in a total of $235 + 2350 = 2585$ data points.

Additionally, every MRI scan of all four image modalities is standardized such that

$$s_{std}[x, y, z] = s_{orig}[x, y, z] - \frac{1}{N} \sum_{w=0}^{W-1} \sum_{h=0}^{H-1} \sum_{d=0}^{D-1} s_{orig}[w, h, d] \qquad (1)$$

where $x, y, z$ indicates the position of the current voxel in the MRI scan $s$, $W$ is the width of the voxel, $H$ the height of the voxel, $D$ the depth of the voxel, and $N = W \times H \times D$.

For certain CNN structures, the data are additionally scaled down to half of its size to improve training performance and memory management, such that

$$s_{scaled}[x, y, z] = \frac{1}{8} \sum_{w=2x}^{2x+1} \sum_{h=2y}^{2y+1} \sum_{d=2z}^{2z+1} s_{std}[w, h, d] \qquad (2)$$

with $0 \leq x \leq \lfloor W/2 \rfloor, 0 \leq y \leq \lfloor H/2 \rfloor, 0 \leq z \leq \lfloor D/2 \rfloor$. To see for which designs the scaling was applied and how it affected the results, we refer to Section 3. Figure 2 shows an exemplary scan after each processing step. The scans shown are centered cross-sections of patient no. 365 of the BraTS2020 dataset with three different views: axial, sagittal, and coronal. The histogram shows relative color changes. After rotation, no changes are visible. The standardization shifts the image towards the center, which is visualized in the example image as a slightly darker scan. The standard size of a 3D MRI scan in the BraTS2020 dataset is $240 \times 240 \times 155$ pixels. After scaling, the number of pixels and thus the single batch size are reduced to $120 \times 120 \times 78$ pixels.

### 2.2. CNN Structure and Preparation

The CNN structure is based on an implementation by Gupta and Jindal [77], which uses slices of MRI scans and a tumor segmentation for survival prediction [78]. The implementation of the CNN was performed in python using TensorFlow and Keras. To use the whole 3D voxel for training and optimizing the performance of the network, additional

changes in network size, layers, and input sizes were required. Furthermore, instead of training the network on selected slices of different MRI sequences, the CNN is trained only on the 3D scaled type but therefore uses the whole 3D voxel as input. The overall network structure can be seen in Figure 3. A total of five convolutional layers with different numbers of feature layers were used to extract meaningful features. Additionally, the age of the patients was considered, as this has already been proven to be a good indicator for survival prediction [35]. The output of the neural network is the days of survival from a given patient, normalized in the range of $[0, 1]$ with $1 = max\_days$, while $max\_days$ is derived from the BraTS dataset and is the longest surviving patient. For the loss function, the mean squared error (MSE) is used as well as the Adam optimizer.

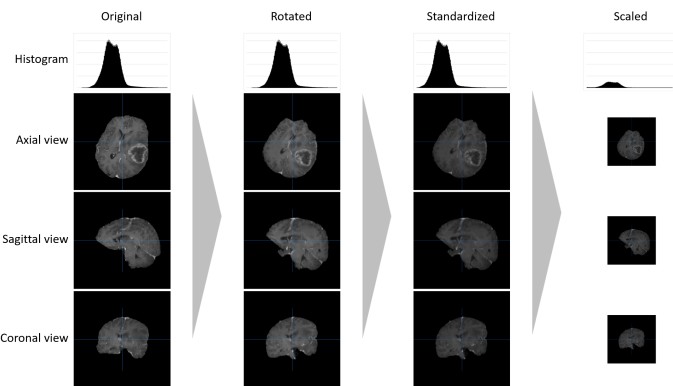

**Figure 2.** The pre-processing steps are conducted to increase the data size and optimize performance.

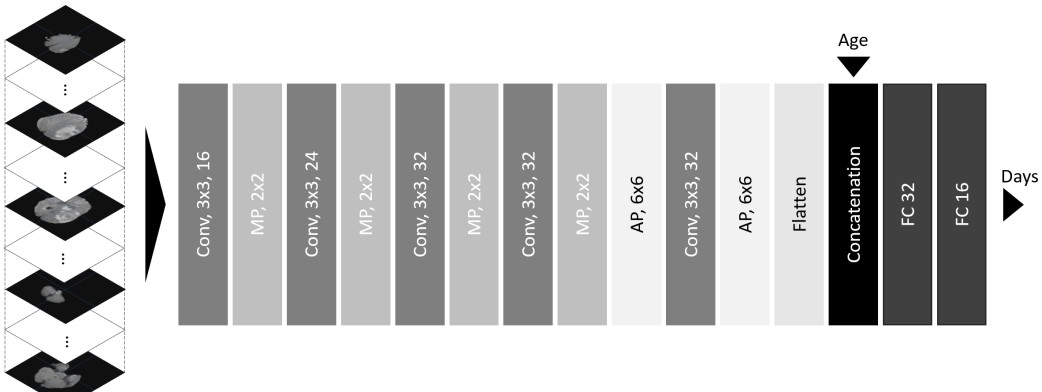

**Figure 3.** Structure of the designed CNN for training 3D voxels from the MRI images as input, based on [78]. The network consists of different layers, such as convolutional (Conv), max pooling (MP), average pooling (AP), concatenation, and fully connected layers (FC).

To train the CNN, the pre-processed data, as discussed in Section 2.1, are split into training and test sets (80/20). Training was conducted on three different network designs: one training was performed using the full-sized voxel and two-dimensional convolutional layers, using 2D feature detectors. Another CNN was trained on 2D convolutional layers as well but used scaled scans as the input, as discussed in Section 2.1. The third CNN structure used 3D feature detectors and the scaled MRI scan as input.

For training, the workload was outsourced to AWS, specifically to ec2 instances of the g4dn family, which is a product group that has up to 4 GPUs (NVIDIA T4 GPUs). As for the machine image, we chose Amazon's deep learning AMI based on Ubuntu 18.04 because it had a lot of the requirements already preinstalled and all issues with GPU drivers already solved. Our augmented dataset was uploaded to a private S3 bucket so the instances could access them via Amazon's in-house connection. The training was performed in parallel with the cheapest instance (g4dn.xlarge), which allowed us to simultaneously train up to 5 different networks.

### 2.3. Explainability

To explain the output of the CNN, we evaluated different pre-existing libraries and selected SHAP [28], available via https://github.com/slundberg/shap (accessed on 6 June 2021). SHAP is a XAI framework that provides, among other features, visual interpretation for a given model on a global level [73]. Additionally, SHAP contains an optimized explainer for deep models.

For each input feature, SHAP assigns a value of how important it was for the output [28]. To calculate these importance values, it offers different calculation methods, including two model agnostic methods that can be applied regardless of the type of trained network and four specific model methods, one being DeepExplainer. In this work, DeepExplainer is used to identify the importance values for a given input combination of 3D MRI voxel and age value. DeepExplainer leverages the possibility to efficiently approximate SHAP values for a deep neural network model by recursively passing DeepLIFT multipliers, as described in [70], deriving an effective linearization technique from the SHAP values. This avoids the need to heuristically choose a linearization method. By passing an example data point as input for DeepExplainer, it determines the importance values for every pixel in the 3D voxel, as well as for the age value. This importance value can then be visualized accordingly by integrating them into a background image, which represents the input. A visualization example is given in Figure 4. Furthermore, DeepExplainer also implements enhancements over the original DeepLIFT implementation with, e.g., the usage of the Shapely equation to linearize components [70].

In Figure 4, a slice of the MRI can be seen before and after it is overlayed with SHAP values to explain the prediction of a patient's survival rate. The SHAP values show the influence of each scan voxel on the output value and indicate an increase (red color) or decrease in survival (blue color). Thus, the predicted survival rate of the patient can be represented as the sum of SHAP values over all voxels. Particularly interesting is that this patient already had a partially resected tumor, and the CNN placed high importance on the area where the tumor was removed.

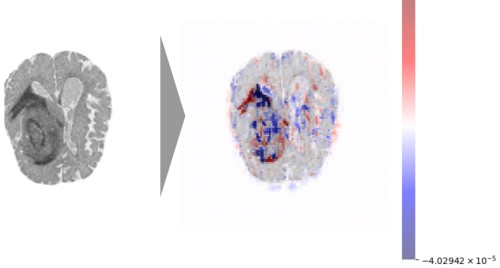

**Figure 4.** Example image of an MRI scan without and with overlayed values from SHAP's DeepExplainer. Red colors indicate an increase, and blue colors a decrease in the network's output value, which is in the range $[0, 1]$ and represents the predicted survival rate. The image shows part of a FLAIR scan from patient no 338. Predicted survival rate: 55 days, actual survival rate: 80 days.

### 3. Evaluation and Results

In this section, we present results from our trained networks and compare model performances. First, a CNN is trained on two different datasets: the original BraTS2020 dataset containing 235 data points and the pre-processed dataset, which contains a total of 2585 data points. A comparison of these results is shown in Table 1. For a statistical analysis on the used dataset, we refer to [75]. Then, the pre-processed dataset is used to compare three different network models with different image sizes and feature detectors. Two networks were trained using 2D feature detectors on a full-size scan (2D full size), as well as on a scaled scan (2D scaled). The third network was trained using 3D feature detectors on a scaled scan (3D scaled).

All four given MRI sequences (t1, t1ce, t2, flair) were trained on the pre-processed dataset on different network structures (see Section 2.2) to compare their overall perfor-

mance. As the sizes of the network and input scans differ, it was also necessary to adopt other training-specific parameters, such as batch size, epoch size, and samples per iteration. Table 2 gives an overview of the used parameters.

**Table 1.** Performance comparison of a trained network on t1 MRI scans with and without pre-processed datasets.

| Dataset Type | Evaluation Metrics (Test Set) | | | | |
|---|---|---|---|---|---|
| | Acc. (%) | MSE (*d*) | Median SE (*d*) | stdSE (*d*) | SpearmanR ($\rho$) |
| original | 57.1 | 127,576.69 | 38,449.52 | 248,209.82 | 0.252 |
| pre-processed | 94.0 | 19,370.85 | 2310.53 | 68,774.36 | 0.934 |

For the evaluation, the accuracy of the network was calculated by assigning every output one out of three class labels, as defined in the BraTS challenge [2,75,76]. Class 0 is assigned for a predicted survival rate of under ten months ($d < 300$), class 1 if the output is within 10–15 months ($300 \leq d \leq 450$), and class 2 if the survival is longer than 15 months ($d > 450$). The accuracy is then calculated as the ratio of the correctly predicted survival class ($T_0, T_1, T_2$) over all predictions, including false classifications ($F_0, F_1, F_2$)

$$Accuracy = \frac{T_0 + T_1 + T_2}{T_0 + F_0 + T_1 + F_1 + T_2 + F_2} * 100 \tag{3}$$

Additionally, other squared error metrics such as the mean squared error (MSE), the median squared error, and the standard deviation of the squared error (stdSE), all given in days $d$, were calculated together with Spearman's rank correlation coefficient [79] (stated as $\rho$).

**Table 2.** Specific parameters for the three different network structures used, including input size, feature detector (FD), batch size, epoch size, number of samples per iteration and total number of iterations.

| CNN Definition | Input Size ($w \times h \times d + age$) | FD | Batch Size | Epoch Size | Samples Per Iteration | Total Iterations |
|---|---|---|---|---|---|---|
| 2D full size | $240 \times 240 \times 155 + 1$ | 2D | 32 | 100 | 50 | 13,000 |
| 2D scaled | $120 \times 120 \times 78 + 1$ | 2D | 64 | 400 | 400 | 23,000 |
| 3D scaled | $120 \times 120 \times 78 + 1$ | 3D | 16 | 100 | 50 | 6000 |

Figure 5 shows the loss functions of different network structures during training. The peaks indicate the overall loss on the network, while the individual valleys occur due to the batch size and the fast convergence of the batches during training. Figure 5a shows a non-converging loss curve, indicating that the network does not find the required features from the input setting. To overcome this issue, the input data were scaled down to improve the extraction of the CNN feature detectors (Figure 5b). Figure 5c shows the loss curve on the network with a 3D feature detector. While this network design converges faster in terms of epochs, the total training time is increased (see Table 3).

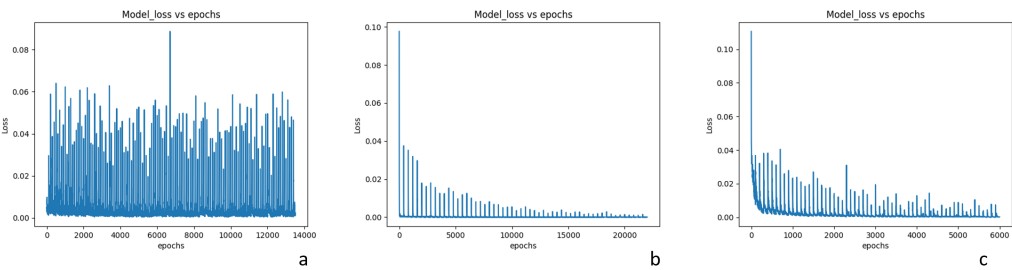

**Figure 5.** Example visualization of the loss functions of the different network structures for training on t2 MRI scans. (**a**) 2D full size (**b**) 2D scaled (**c**) 3D scaled.

**Table 3.** Overview of different CNN test results given different evaluation metrics.

| CNN Type | MRI Sequence | Time (min) | Evaluation Metrics (Test Set) | | | | |
|---|---|---|---|---|---|---|---|
| | | | Acc. (%) | MSE (*d*) | Median SE (*d*) | stdSE (*d*) | SpearmanR ($\rho$) |
| 2D full size | T1 | 537 | 42.0 | 152,284.23 | 61,174.52 | 225,719.97 | 0.329 |
| 2D full size | T1CE | 499 | 43.5 | 183,690.40 | 30,002.06 | 389,383.37 | 0.383 |
| 2D full size | T2 | 523 | 41.3 | 183,086 | 53,753.30 | 322,327.76 | 0.260 |
| 2D full size | FLAIR | 518 | 44.7 | 171,695.34 | 29,713.50 | 322,903.01 | 0.337 |
| 2D scaled | T1 | 297 | 94.0 | 19,370.85 | 2310.53 | 68,774.36 | 0.934 |
| 2D scaled | T1CE | 260 | 84.8 | 15,181.36 | 2323.54 | 57 120.51 | 0.948 |
| 2D scaled | T2 | 252 | 86.2 | 13,362.36 | 1435.69 | 49,180.29 | 0.951 |
| 2D scaled | FLAIR | 266 | 87.8 | 12,619.52 | 1722.33 | 46,454.14 | 0.974 |
| 3D scaled | T1 | 832 | 73.7 | 22,853.60 | 3955.39 | 83 608.18 | 0.874 |
| 3D scaled | T1CE | 829 | 79.8 | 35,815.96 | 2771.14 | 131,752.81 | 0.957 |
| 3D scaled | T2 | 825 | 89.4 | 19,423.55 | 1672.75 | 74,759.27 | 0.950 |
| 3D scaled | FLAIR | 823 | 81.2 | 16,603.59 | 2620.99 | 60,487.03 | 0.942 |

Figures 6 and 7 present two exemplary samples with SHAP visualizations. Thereby, we display visualizations based on FLAIR images using the "2D full size" model, which is also the best performing of all compared networks according to validation results, as shown in Table 4. The model shows lower accuracies under 50% when using other image modalities than FLAIR. Input images are shown as nearly transparent grayscale. The color indicates pixels affecting the model output. The color code (red, blue) defines a change in the model output value, which is pushed between the minimum and maximum survival of samples between 0 and 788 days.

For the evaluation of the test set, additional metrics on class accuracies are shown in Table 5, which are not provided for the validation set by the MICCAI-BraTS 2020 leaderboard. The labels in the table denote the true/false class assignments, as described in Equation (3).

Training time differs in every network structure. The network "2D scaled" is the fastest, allowing fast training results. Furthermore, the training accuracy in 2D scaled is rather high, ranging from 84.8% (t1ce) to 94.0% (t1).

For validation of the network, the validation set of the BraTS2020 data is used. The results are shown in Table 4. The results were calculated on the CBICA Image Processing Portal (https://ipp.cbica.upenn.edu/ (accessed on 23 November 2021)), which compares the results to the non-public validation set for the MICCAI-BraTS2020 Validation Survival Leaderboard (https://www.cbica.upenn.edu/BraTS20/lboardValidationSurvival.html (accessed on 21 March 2022)).

Validation is based on ground truth labels by expert board-certified neuroradiologists and was not provided to the participants of the BraTS challenge directly. As the labels of the validation set are unknown, the metrics could not be calculated by the authors, and the results for the validation set could not be verified. Some results seem to be very similar or questionable (marked with *). For example, in the calculation of the overall accuracy, the results were the same when being uploaded to the portal at the same time. However, while the exact root cause of similarities in the accuracy of validation results, as shown in Table 4, is unclear, the authors assume they occur due to the small size of the provided validation data set of 29 patients. A summary of performances of the 2D full size and 3D scaled models based on FLAIR images is presented by scatter-plots comparing prediction over ground-truth in Figure 8. Further plots on the various models and all imaging modalities can be found in the "fig" folder of the repository on https://gitlab.com/matte3000/xai-for-brain-img-surv/ (created 29 April 2021, last updated 31 August 2022). The scatter-plots highlight a good correlation between predicted and provided survival rates in the case of the 3D scaled network for all classes (b). Whereas for the 2D full-size model performance, distinction between classes exhibits greater divergence.

**Table 4.** Overview of different CNN validation results given different evaluation metrics. The results are taken from the MICCAI-BraTS 2020 leaderboard. As the metrics were not calculated by the authors, the results that were suspiciously similar or questionable have been marked with *.

| CNN Type | MRI Sequence | Evaluation Metrics (Test Set) | | | | |
|---|---|---|---|---|---|---|
| | | Acc. (%) | MSE ($d$) | Median SE ($d$) | stdSE ($d$) | SpearmanR ($\rho$) |
| 2D full size | T1 | 44.8% * | 113,420.55 | 65,536.00 | 146,473.46 | 0.267 |
| 2D full size | T1CE | 44.8% * | 127,564.79 | 20,449.00 | 197,625.03 | 0.324 |
| 2D full size | T2 | 37.9% | 147,032.52 | 69,169.00 | 211,555.71 | 0.132 |
| 2D full size | FLAIR | 55.2% | 69,941.35 | 12,769.00 | 116,749.59 | 0.435 |
| 2D scaled | T1 | 48.3% * | 135,167.90 | 33,856.00 | 194,580.94 | 0.024 * |
| 2D scaled | T1CE | 48.3% * | 94,662.03 | 41,616.00 | 153,731.94 | 0.352 |
| 2D scaled | T2 | 48.8% * | 94,288.35 | 18,225.00 | 147,035.02 | 0.218 |
| 2D scaled | FLAIR | 31.0% | 119,759.07 | 44,944.00 | 176,681.82 | 0.184 |
| 3D scaled | T1 | 44.8% * | 90,073.24 | 44,100.00 | 154,947.23 | 0.249 |
| 3D scaled | T1CE | 48.3% * | 105,020.93 | 25,600.00 | 184,385.83 | 0.270 |
| 3D scaled | T2 | 44.8% * | 130,643.76 | 33,856.00 | 170,256.56 | 0.134 |
| 3D scaled | FLAIR | 31.0% | 111,939.31 | 45,796.00 | 164,669.90 | −0.020 * |

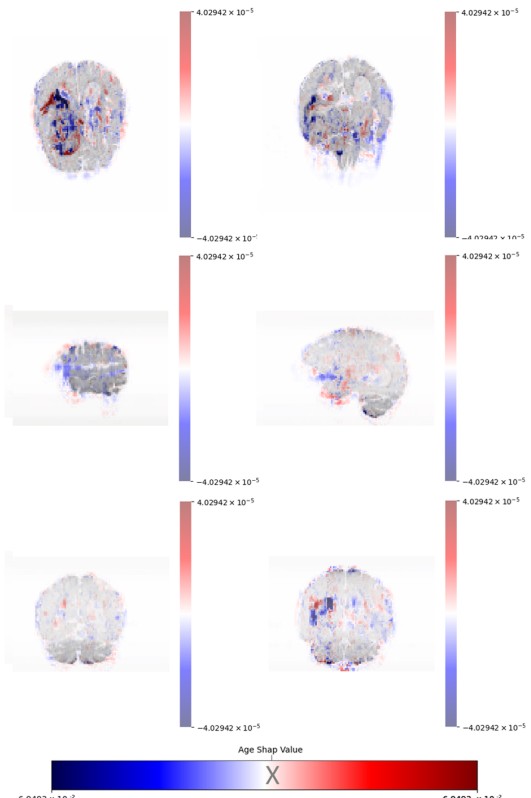

**Figure 6.** Example visualization of patient 338's (sub total resection) SHAP values from network "2D full size", trained with FLAIR scans from different angles (top down: axial, sagittal, coronal). The SHAP features focus on the resected area but also show minor phantom features. Age does not have any major influence on the predicted survival rate. Red SHAP values indicate an increase, and blue values indicate a decrease in the network's output value, which is in the range $[0,1]$ and represents the predicted survival rate. Predicted survival rate: 55 days, actual survival rate: 80 days.

**Table 5.** Additional evaluation metrics on the class accuracies of the test dataset for the three survival rate classes. $T_i$ describes the number of correct labels assigned to class $i$, $F_i$ the number of incorrect labels in class $i$. $Acc_i$ describes the class accuracy of class $i$, while $i = 0$ for $d < 300$, $i = 1$ for $300 \leq d \leq 450$), $i = 2$ for $d > 450$ ($d =$ days).

| CNN Type | MRI Sequence | Evaluation Metrics (Test Set) | | | | | | | | |
|---|---|---|---|---|---|---|---|---|---|---|
| | | $T_0$ | $F_0$ | $Acc_0$ (%) | $T_1$ | $F_1$ | $Acc_1$ (%) | $T_2$ | $F_2$ | $Acc_2$ (%) |
| 2D full size | T1 | 46 | 27 | 63.0 | 36 | 95 | 27.5 | 153 | 202 | 43.1 |
| 2D full size | T1CE | 158 | 205 | 43.5 | 42 | 87 | 32.5 | 43 | 24 | 64.2 |
| 2D full size | T2 | 82 | 71 | 53.4 | 38 | 102 | 27.1 | 111 | 155 | 41.7 |
| 2D full size | FLAIR | 112 | 112 | 50.0 | 179 | 119 | 33.2 | 78 | 78 | 50.0 |
| 2D scaled | T1 | 179 | 32 | 84.8 | 129 | 42 | 75.4 | 170 | 7 | 96.0 |
| 2D scaled | T1CE | 179 | 33 | 84.4 | 128 | 45 | 74.0 | 167 | 7 | 96.0 |
| 2D scaled | T2 | 177 | 24 | 88.1 | 140 | 47 | 74.9 | 165 | 6 | 96.5 |
| 2D scaled | FLAIR | 192 | 35 | 84.6 | 130 | 31 | 80.7 | 169 | 2 | 98.8 |
| 3D scaled | T1 | 156 | 19 | 89.1 | 79 | 49 | 61.7 | 177 | 79 | 69.1 |
| 3D scaled | T1CE | 194 | 51 | 79.2 | 117 | 62 | 65.4 | 135 | 0 | 100.0 |
| 3D scaled | T2 | 177 | 12 | 93.7 | 139 | 32 | 81.3 | 184 | 15 | 92.5 |
| 3D scaled | FLAIR | 194 | 63 | 75.5 | 107 | 39 | 73.3 | 153 | 3 | 98.1 |

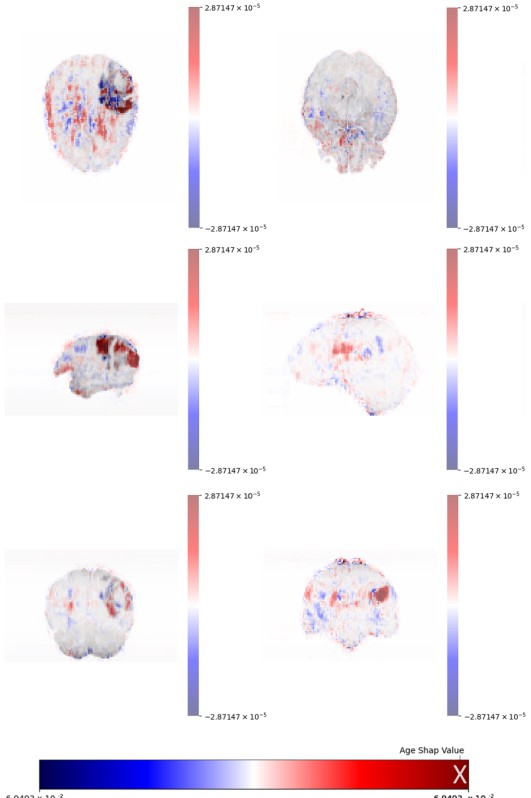

**Figure 7.** Example visualization of patient 004's (gross total resection) SHAP values from network "2D full size", trained with FLAIR scans from different angles (top down: axial, sagittal, coronal). Red SHAP values indicate an increase, and blue values indicate a decrease in the network's output value, which is in the range $[0, 1]$ and represents the predicted survival rate. Here, the focus on the tumor region can be seen nicely, indicating that many features for predicting the survival rate come from this region. Age seems to have a major impact on the patient's survival prediction. Predicted survival rate: 621 days, actual survival rate: 788 days.

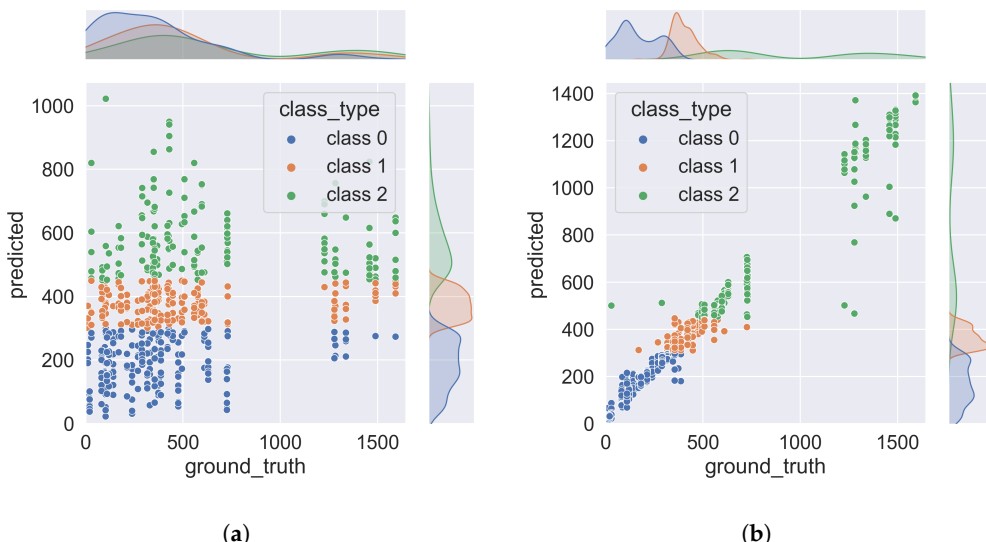

(**a**)            (**b**)

**Figure 8.** Scatter-plots comparing prediction vs. ground-truth of (**a**) 2D full size and (**b**) 3D scaled networks trained on FLAIR images, individual class performances for class 0 corresponds to $d < 300$, class 1 for $300 \leq d \leq 450$), class 2 for $d > 450$ ($d =$ days).

## 4. Discussion

In this paper, we discuss model performances of various networks and suggest XAI as an add-on to support the evaluation process.

Generally, depending on the application and method of radiomics data, as well as the type or amount of data used, there are varying error rates in reporting. Furthermore, depending on the task/application, the error type differs. While, for example, a dice coefficient might be the desired metric to gauge the similarity of two samples in the case of an image segmentation task, the class accuracy will be more informative for classification tasks. For regression tasks, on the other hand, squared errors are a common metric. Tolerable errors for the different applications differ substantially and change frequently in the field of radiomics. For the task of survival prediction, we refer to the MICCAI-BraTS Leaderboard (https://www.cbica.upenn.edu/BraTS20/lboardValidationSurvival.html (accessed on 21 March 2022)), which lists the results for predicting the survival rate by multiple teams during the multimodal brain tumor segmentation challenge.

In 2D full size (Table 3), the training accuracy and SpearmanR are rather low, and the other metrics are rather high, indicating that the features were not detected correctly and that the network did not converge. The results compared to other network structures and the loss function from Figure 5 might indicate that this network does not perform as well as the others. This is in line with the comparison of predicted versus real values presented in Figure 8. An adaption of the network (e.g., feature detectors) might be considered to solve this issue. However, as the focus of this work is not on the optimization of the network structure but on the support of interpretability of the network using SHAP features, no optimization is conducted.

Data for the BraTS challenge are harmonized by the providers in order to be comparable. The selection of the class distributions on the training and test dataset was conducted as suggested by the data providers, who derived thresholds after statistical consideration of the survival distributions across the complete dataset based on equal quantiles from the median overall survival to avoid potential bias toward one of the survival classes [76].

Additional metrics and the class accuracy of all network types (see Table 5) show a good accuracy distribution of all three classes. Only the trained network models "2D full size" show a drop in accuracy over the classes. For example, "2D full size" of type T1 has a class accuracy of 27.5% in class 1, which is lower than class 0 (63.0%) and class 2 (43.1%), which is similarly true for "2D full size" of type T2 and T1CE to a slightly lesser extent. Models based on scaled input exhibit more even accuracies among classes.

On the other hand, when comparing the validation results of the best fitting networks with the training results, the accuracies differ a lot, indicating an overfitted network. This indicates that the network does not train the features to predict the survival rate as well as expected, given the training results. As mentioned above, we did not optimize the network structure but used this as a baseline for the interpretation support using Shapley overlays. Still, when evaluating the network performances on the validation set (Table 4), the network "2D full size" trained on flair images resulted in the best score and also indicated good performance when compared to the tied winners of the BraTS2020 survival challenge [80,81].

These results show the necessity of additional data sets for network validation. However, since only a limited number of datasets are available in the followed domain, every additional dataset that can be used for training is of great importance. To be able to evaluate a network despite the lack of validation data, we use SHAP features, which allow for the interpretation of the trained network.

For this, we use the results from Table 4 as a consulting factor when evaluating the networks using SHAP features. Without the validation sets, one might conclude that the "2D scaled" network structure yields the best results and discard the "2D full size" network. However, when using XAI to extract interpretations of the trained network, similar conclusions can be drawn when a validation set is available. Using SHAP to visualize the decision on the trained CNN regression, one can compare and verify the performance by evaluating the visualizations of a few labeled patient data sets.

For example, Figure 6 shows the visualization of a subtotal resection (STR) data point (patient 338) on the "2D full size" network structure, trained on flair scans. Blue colors indicate a reduction in the patient's life expectancy, while red colors indicate an increase in the survival rate. The predicted survival rate was 55 days, whereas the actual survival was 80 days. One can observe that the SHAP features used to estimate the survival of the patient focus on the region where the tumor was partially resected. In other regions, various small flocks are highlighted, which indicate noise of the trained CNN that seems to have only a minor influence on the result. The influence of age is negligible in the patient's survival prediction. Another example using a gross total resection (GTR) data sample from patient 004 on the "2D full size" network, trained on flair scans, can be seen in Figure 7. The predicted survival rate was 621 days vs. 788 actual days. Here again, the influence of the tumor environment on the survival prediction can be seen clearly, as the SHAP features focus mainly on the tumor resection. In this sample, however, the survival prediction is highly influenced by the patient's age.

The existence of a visualization of the output of a network supports the evaluation of the model design and the quality of the output with the help of expert knowledge. For example, as explained in the examples above, the focus of the SHAP features on relevant locations indicates that the network can recognize the tumor region well. In addition, flocks that are directly unrelated to the tumor are also highlighted. This can now be detected using visual support and does not require additional extraction of information through more data.

For further analysis of the quality of the SHAP features, a comparison with the provided segmentation of the BraTS2020 dataset was conducted. Therefore, the SHAP result was visualized together with the raw MRI scan as well as the segmentation data, as shown in Figure 9. The annotations of the tumor segmentation comprise the necrotic and non-enhancing tumor core (dark grey), the GD-enhancing tumor (white), and the peritumoral edema (light grey), as presented by [76]. When comparing the learned features of the network to the raw FLAIR MRI scan of patient 004, a clear focus on the tumor's resection area can be seen, which influences the survival rate in both directions with a focus on positive features, indicating a longer survival rate. Additionally, the peritumoral edema (light grey) provides the most information on the survival rate, while the necrotic core (dark grey) has only little influence. When compared to an example slice of patient 338 (see Figure 10), who had a significantly shorter survival rate, a similar focus on the peritumoral

edema can be seen—this time, however, with a more negative impact. Additionally, a negative influence on the survival rate can be seen on the edges between the necrotic core (dark grey) and the GD-enhancing tumor (white). While the negative influence of the necrotic core correlates with medical findings to negatively influence a patient's survival rate, the impact of the peritumoral edema cannot be identified as it seems to have a significant positive and negative impact on the survival rate.

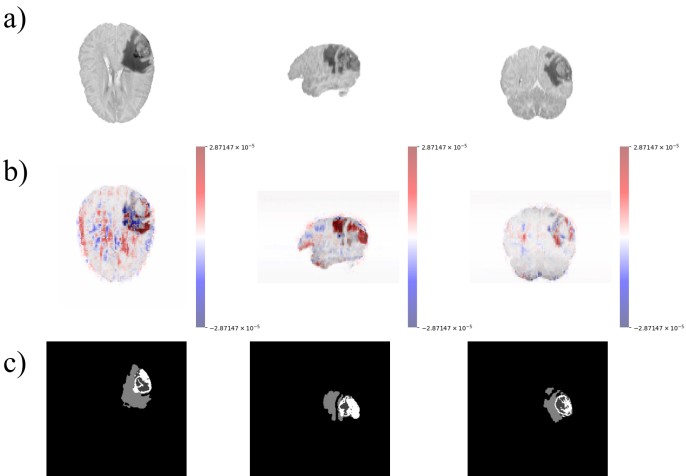

**Figure 9.** Example analysis of the SHAP features. (**a**) The reference FLAIR image of patient 004, (**b**) the SHAP values and impact on the predicted survival rate from network "2D full size", (**c**) the reference tumor segmentation, as provided in the BraTS2020 dataset [76]. Predicted survival rate: 621 days, actual survival rate: 788 days.

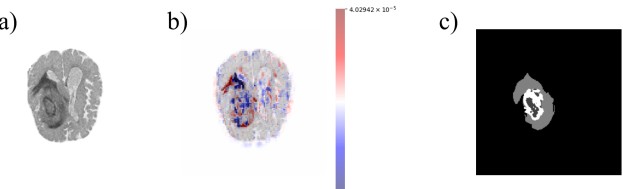

**Figure 10.** Example slice for the analysis of SHAP features. (**a**) The reference FLAIR image of patient 338, (**b**) the SHAP values and impact on the predicted survival rate from network "2D full size", (**c**) the reference tumor segmentation as provided in the BraTS2020 dataset [76]. Predicted survival rate: 55 days, actual survival rate: 80 days.

While research usually focuses on the validation using additional metrics and data, our results suggest that XAI can support the decision based on already used data sets and provide additional information for a domain expert to evaluate the network. Having a visualization of SHAP features thus helps to evaluate the overall quality of the trained network and supports the decision-making process regarding its applicability to survival prediction.

While the proposed approach improves the efficient use of data and the evaluation of a network, it also requires knowledge of the training domain to assess the XAI approach. In the domain of survival prediction on MRI scans, information on the patient's glioma (e.g., size, resections status, location) has to be provided for the verification of SHAP features.

There are common observations familiar to neuroradiologists described in VASARI (Visually AcceSAble Rembrandt Images) as well as other radiomics features [82]. An interesting future research task would be to correlate SHAP overlays with a subset of these features.

Radiomics is said to improve diagnosis rather than replace radiologists [5]. Future work may involve further causability studies: explaining a patient's prediction can be of particular interest for comparison with a diagnosis by an experienced radiologist and could be further studied for novel feature detection.

## 5. Conclusions

In this study, we presented an application of visual explanations to interpret network models that were trained to predict the survival rate of glioma patients.

We compared three different networks based on full size or scaled input, resulting in accuracies between 31 and 55.2%, with a mean individual class accuracy of around 71%. The model "2D full size" trained on flair images yields the highest accuracy of all tested models presenting good performance compared to winners of the BraTS2020 survival challenge.

The analysis shows that SHAP features can support the interpretation of training results and shows that even if high accuracy is achieved, the network might still be trained poorly. This, in turn, helps to evaluate a network model and to optimize it, if necessary, to improve predictions. Additionally, the pre-processing of the dataset allows for bigger training sets, which can help to improve the performance of the network.

The main limitation of the proposed model is defined by the lack of additional data sets, in particular for network validation. This manuscript is intended to focus on the use of explainability for interpreting the results of radiomic models for survival prediction in glioma at the expense of network optimization.

In cases where only limited training data are available, SHAP can play an important role in conjunction with pre-processing steps to evaluate trained network models. Future studies may use this knowledge to improve the CNN structure and results to optimize network performance.

For future research tasks, limitations of the BraTS2020 dataset for the task of survival prediction may be further explored, such as the unequal distribution of the different grades as well as HGG/LGG differences [83]. For this case, other datasets exist but are yet to be harmonized [4]. Follow-up work will include model refinement by integrating heterogeneous data from multiple sources. Better-performing networks can be further used to investigate SHAP interpretations based on different image modalities.

**Author Contributions:** C.J.-Q. and F.J. participated in the conceptualization. M.E., E.M., A.H., C.J.-Q. and F.J. wrote the main manuscript text and participated in formal analysis. M.E. and E.M. implemented the method and prepared Figures 2–9. C.J.-Q. and F.J. prepared Figure 1. M.E. participated with the MICCAI-BraTS 2020 Leaderboard. All authors have read and agreed to the published version of the manuscript.

**Funding:** M.E., E.M., C.J.-Q., and F.J. had no role in funding. A.H. received partial funding from the Austrian Science Fund (FWF), Project: P-32554 explainable Artificial Intelligence.

**Institutional Review Board Statement:** Not applicable.

**Informed Consent Statement:** Not applicable.

**Data Availability Statement:** The implementation, source code, results, and trained models are available on https://gitlab.com/matte3000/xai-for-brain-img-surv (created 29 April 2021, last updated 31 August 2022)).

**Acknowledgments:** We thank all the data providers. We dedicate our work in memoriam to our family members and friends we have lost. If we may contribute even tiny steps to help to save lives in the future, our mission was worth our passion, enthusiasm and effort. Please visit our project homepage, available online at: https://human-centered.ai/project/tugrovis (accessed on 23 November 2021).

**Conflicts of Interest:** The authors declare no conflict of interest.

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
