# Peer review of "Interpretable Machine Learning with Brain Image and Survival Data"

_biomedinformatics, doi:10.3390/biomedinformatics2030031_

Round 1
Reviewer 1 Report
Paper deals with important task. The authors tryied to developed an explainable AI to interpret the decision-making of an ML algorithm on the use case of predicting the survival rate of patients with brain tumors based on MRI scans
Suggestions:
1. In the abstract section I found “in this thesis”. But this is a paper....
2. The first main contribution is unclear. This proposal “the proposal of a pre-processing step to augment the used dataset and increase the number of data samples for training” is not new. In addition use the such a simple procedure as a rotation is not good in mant cases. Please clearly fromalate your contribution and argued the cjhosed methodology.
3. The authors used 235 images and augmented it. I cant find the information about size of the training sample
4. It would be good to see the experiments with othe, more bigestt dataset as deep learning should be used for a Big Data but no small data as in your case
5. The introduction section should be significantly extended.
6. The conclusion section should be extended using: 1) numerical results obtained in the paper; 2) limitations of the proposed approach; 3) prospects for future research.
7. Some of references are outdated. Please fix it using 3-5 years old papers in high-impact journals.
Author Response
We would like to thank you and are grateful for your thorough review of our manuscript.
Ad. 1. In the abstract section I found “in this thesis”. But this is a paper....
Thank you for this comment. We corrected the term.
Ad 2. The first main contribution is unclear. This proposal “the proposal of a pre-processing step to augment the used dataset and increase the number of data samples for training” is not new. In addition use the such a simple procedure as a rotation is not good in mant cases. Please clearly fromalate your contribution and argued the cjhosed methodology.
We refined the description of main contributions accordingly, you can follow our changes highlighted in red.
Ad 3. The authors used 235 images and augmented it. I cant find the information about size of the training sample
Thank you for the question. We already wrote about the resulting data points in the subsection about data preprocessing, but added additional information on the original dataset: We used the BraTS 2020 dataset that comprises 369 training samples that comprise 235 patients with either HGG or LGG and survival data, while we filtered on those 235 actual cases that have survival data. These were further augmented by random rotation, resulting in a total of 2585 data, consisting of 2350 newly rotated samples additional to the 235 original ones. To train the CNN, the pre-processed data is then split into training and test sets (80/20). Therefore, the sample size was 2068.
The size of a MRI input voxel for the network varies over the network type as it was scaled. The different 3D voxel sizes which were used for the network can be seen in Table 1.
Ad 4. It would be good to see the experiments with othe, more bigestt dataset as deep learning should be used for a Big Data but no small data as in your case
In this work we concentrated on the provided data from BraTS, we discuss this point as limitation and prospect to future work.
Ad 5. The introduction section should be significantly extended.
We concentrated on pointing out some of the interesting works on glioma survival prediction with methodical approaches to survival prediction, further background on MRI Regression/Classification and up to introducing the topic of CNNs to XAI in MRI Imaging. We added only some more information, again reviewed that manuscript and compared it with other papers in the field. We believe, the introduction with the additional background description structure into the three given subsections (page1-4) is comprehensible enough to scientists working outside the topic of the paper but also concise enough to include the important terms. While this manuscript is not a review article, adding further information seems out of scope of a research article.
Ad 6. The conclusion section should be extended using: 1) numerical results obtained in the paper; 2) limitations of the proposed approach; 3) prospects for future research.
We extended the section accordingly.
Ad 7. Some of references are outdated. Please fix it using 3-5 years old papers in high-impact journals.
Thank you for this comment. We corrected several former arxiv references where we found updated publications as well as checked on more qualitative and fitting ones.
Reviewer 2 Report
In this study, the authors provide an approach to interpret network models which were trained to predict the survival rate of glioma patients. SHAP features can support the interpretation high accuracy can still reflect poor model training. Moreover, the pre-processing the datasets improves the performance of the network. The work shows promise in the field of AI and DNNs. However, I do have some questions:
1) Could you comment on the errors in reporting radiomics data and how much of the data-error is acceptable for training the model?
2) Does the model account for mislabeling of a data (Flase positives and False Negatives).
3) Authors mentioned that DNNs in general are considered as black boxes. This cause problems with reproducibility. The study does not really address this limitation.
4) Was there any skewness observed in the data? What kind of negative sets were used, if any, to account for the skewness?
5) Some more statistical analysis needs to be provided. There has to be something that shows the results you obtain are not random or by chance.
Author Response
We would like to thank you and are grateful for your thorough review of our manuscript.
Ad. 1) Could you comment on the errors in reporting radiomics data and how much of the data-error is acceptable for training the model?
Depending on the application and used method of radiomics data, as well as the data used (type, amount etc.), the error rates differ. Also, depending on the task/application, the error type differs. While for example for an image segmentation task a dice coefficient might be the desired metric to gauge the similarity of two samples, for classification tasks it can be the class accuracy. For regression tasks on the other hand, squared errors (MSE, RMSE) are a common metric. The “acceptable” errors for the different applications differ a lot and change frequently in the field of radiomics. For the task of survival prediction, we refer to the MICCAI-BraTS Leaderboard which states the results for predicting the survival rate on the BraTS2020 dataset, which was used by multiple teams during the multimodal brain tumor segmentation challenge.
Ad. 2) Does the model account for mislabeling of a data (Flase positives and False Negatives).
We did not check on mislabeled data since we used the challenge’s data. We added information in the methods section. As stated in conclusion, a further investigation on other data sets is to be conducted as future work.
Ad. 3) Authors mentioned that DNNs in general are considered as black boxes. This cause problems with reproducibility. The study does not really address this limitation.
Thank you for this comment. This is indeed a common problem when using DNNs for prediction. To overcome this, however, we’ve decided to use only publicly available datasets, provide the source code for our proposed method and also provide the trained models which are described in this paper.
Ad. 4) Was there any skewness observed in the data? What kind of negative sets were used, if any, to account for the skewness?
The data for the BraTS challenge is harmonized by the dataset providers to be comparable. The selection of the class distributions on the training and test dataset was conducted as suggested by the dataset providers (see reference [73], Bakas et al. (2018) “Identifying the Best Machine Learning Algorithms for Brain Tumor Segmentation, Progression Assessment, and Overall Survival Prediction in the BRATS Challenge”). To quote these authors “These thresholds were derived after statistical consideration of the survival distributions across the complete dataset. Specifically, we chose these thresholds based on equal quantiles from the median OS (approximately 12.5 months) to avoid potential bias towards one of the survival groups (short- vs long- survivors) and while considering that discrimination of groups should be clinically meaningful.” For further information on the dataset, we refer to the reference above.
Ad. 5) Some more statistical analysis needs to be provided. There has to be something that shows the results you obtain are not random or by chance.
Statistical analysis on the provided dataset is given in the original publication (see reference [73], Bakas et al. (2018) “Identifying the Best Machine Learning Algorithms for Brain Tumor Segmentation, Progression Assessment, and Overall Survival Prediction in the BRATS Challenge”).
To better understand the obtained results on the dataset, we’ve provided additional information on the results, namely the class accuracies, introduced in the new Table 4, and the number of correct/incorrect labels per class, line 300+, and related changes highlighted in red.
Reviewer 3 Report
Review for « Interpretable Machine Learning with brain image and survival Data » in biomedInformatics MDPI
This study uses the MRI scans and clinical information data provided in Brain Tumor Segmentation (BraTS) Challenge 2020 to predict overall patient survival using convolution neural networks (CNN) with three different input data then provide a visual interpretation using shapley additive explanations (SHAP). However, the current status of this study missed some important parts that need to be addressed and needs to be improved to publish.
I have some major concerns about the presentation of the results. Especially claiming SHAP values can be used for evaluation of the model.
Line 56-58-58:
I think, it is a good idea to use interpretability to understand why the predictions were made so by the algorithm however validation cannot rely on extracted features matching the domain knowledge. Prediction performance should be evaluated on a test/validation set.
Table 4: It is not a good idea to include this table in the manuscript as it is, the language used in the paper is also does not sound good. Author should investigate how the values in the table were calculated and calculate the values using their predictions and the provided validation set consists of 29 patients.
I suspect that authors either have an overfitting problem or they are double dipping when splitting the train and test set in the beginning of their training which means they test their algorithm with the data they already have in the training data set.
Either way, most probably overfitting will occur and this risk the learned features generalizability.
294-297: what are similarities mentioned here?
this part is very confusing in general, and lessens the credibility of the proposed method. Needs to be addressed more transparent and clearly.
306-308 and 312-313:
I disagree with the authors using SHAP features on a not optimized model to make sense of it. Any model that wants to present meaningful results must be suitable for the data type, tuned and optimized. Otherwise, if it cannot predict well, the extracted information will also be questionable.
320-322:
I, again, do not think using explainability comes from SHAP feature can be used for validation of the model. Also please report in a figure or table for all or overall SHAP features explains/interprets the prior knowledge.
340-343: Is there a general pattern that SHAPE features overlap with the extracted biological information? How well is this overlap? Could you give an overall statistic or measure it besides prediction accuracy or case-by-case eye-balling evaluation like in the figures given? Please provide the following info in the figures or figure captions: what is the ground truth for the survival and what was predicted.
Quality of the figures are not good and the sizes of the same result figures are not consistent.
369-373: Again, evaluation of a model cannot be done by visualizing the learned features on training data. Authors either should change these claims or evaluate the model’s overall quality by validation and using standard metrics on validation data.
There are some easier to address concerns and here are my suggestions:
Abstract:
There is an abbreviation for artificial intelligence (AI) but not for MRI, BraTS, and SHAP. Make sure to write long versions of the abbreviations where they are used for the first time, especially in the abstract.
Introduction:
Line 54: reference for SHAP is missing here when it is first mentioned.
Line 105, 108, 112, 113, 131, 160, 167:
Sentences start with the reference number, and the numbers are used as a subject of the sentences. I think using passive sentence structure and adding the references at the end of the sentences will improve the writing of the manuscript.
Line 137 -> Line 62
Give long version of XAI earlier in the text when it is first used: move “explainable AI (XAI)” above line 62 from line 137.
Line 142: Causability -> causability
Line 145: What is DIN EN ISO 9241-11? Add a few words of definition to the text.
162: move “SHapley Additive exPlanations (SHAP)” where you first mention SHAP in abstract.
188: What is (r p y)? Add some descriptive text such as, vertical axis (yaw), transverse axis (pitch), longitudinal axis (roll).
208: “the CNN is trained on only one type…” which one?
238 -> 239, no need for new paragraph. And some of these descriptions can also be mentioned in the intro where you introduce your paper’s main points.
239-246: I found this paragraph very difficult to follow. It could be explained step-by-step how the explainability was applied and I think this is the most important part of this article and should be told clearer.
250-251: it would be a good idea to include overall survival prediction in the figures (figure 4, also figure 6-9) since the prediction evaluation is done using overall survival.
253-254: “As described in Figure 2, age is also included as a feature in the training, but the visualization of this feature is not done in SHAP”
I do not see any description mentioned above in figure 2. And please include here, the result for age feature and the reference figure for the visualization of the age feature.
263:
About the performance metrics, what is the algorithms accuracy performances per class, T_0, T_1, and T_3.
Table 2: It is very difficult to understand if a prediction is good or not or compare with another prediction result unless the evaluation metrics are normalized. I suggest authors to report normalized root mean squared error (NMRSE) for the regression problem for example.
Discussion:
---I think per class accuracies are more informative and classification results could be better per class
376-377:
Is it possible to discover new useful information using SHAP features?
Conclusion:
384-385: Do you have any examples/results that SHAPE values are helpful interpreting predictions on validation set? Can they be generalized? Like rule extraction?
394-395: What are the three-class distribution in the training and test data? Did you use stratified sampling? If yes, please report this information and details in the text. If not, it could be the reason for overfitting and poor performance with validation data.
Some references are not up to date, there are arxiv papers from 2017 but they are published in the journals already. Like the most important reference for this paper which is the paper of SHAP (SHapley Additive exPlanations) [60]
Author Response
We would like to thank you and are grateful for your thorough review of our manuscript.
Point 1: Line 56-58-58: I think, it is a good idea to use interpretability to understand why the predictions were made so by the algorithm however validation cannot rely on extracted features matching the domain knowledge. Prediction performance should be evaluated on a test/validation set.
Thank you for this comment, we agree that we have to carefully use shapley values to support the evaluation and not as an independent validation mechanism. We therefore changed our description and discussion to be more comprehensible, throughout all sections, changes highlighted in red.
Please note as summary: We evaluated prediction performance on a test/validation set provided by Brats2020.
Point 2: Table 4: It is not a good idea to include this table in the manuscript as it is, the language used in the paper is also does not sound good. Author should investigate how the values in the table were calculated and calculate the values using their predictions and the provided validation set consists of 29 patients.
Thank you for your comment. In the paper, we describe how the metrics for the experiments are calculated (Line 268-272). For table 4, the “ground truth” labels for the validation dataset are not publicly available and thus cannot be validated by the authors. For clarification, we’ve updated the paper and included this problem verbally.
Please note, we added another table before with details on the class accuracies, so Table 5 represents the former Table 4 now!
Point 3: I suspect that authors either have an overfitting problem or they are double dipping when splitting the train and test set in the beginning of their training which means they test their algorithm with the data they already have in the training data set. Either way, most probably overfitting will occur and this risk the learned features generalizability.
We agree that there is a high probability of overfitting in our algorithm, still the focus of this manuscript is not intended to introduce an optimized accurate model but instead emphasize the addition of explainability methods. As discussed, we focus on shap: The trained network models were used to evaluate the SHAP overlays directly and were not optimized for accuracy. The resulting overfitting of some network structures is therefore seen as a use case of the presented interpretation method.
Point 4: 294-297: what are similarities mentioned here?
We added details to our description in terms of comprehensibility: Similarities in accuracy of T1 and T1CE and T2 approach.
this part is very confusing in general, and lessens the credibility of the proposed method. Needs to be addressed more transparent and clearly.
We extended the paragraph in which similarity is described and added further information on BraTS validation procedures.
Point 5: 306-308 and 312-313: I disagree with the authors using SHAP features on a not optimized model to make sense of it. Any model that wants to present meaningful results must be suitable for the data type, tuned and optimized. Otherwise, if it cannot predict well, the extracted information will also be questionable.
We agree that using an optimized network structure is best for interpreting the results. However, as observed in the work for this manuscript, it is hard to determine when a network structure is optimized enough (could be overfitting as well). In this case, it is useful to use SHAP features to understand the results of the trained model and, if needed, further optimize the model to obtain improved results.
Point 6: 320-322: I, again, do not think using explainability comes from SHAP feature can be used for validation of the model. Also please report in a figure or table for all or overall SHAP features explains/interprets the prior knowledge.
Thank you again for punctuating the term validation. We corrected misunderstandable phrases. Regarding additional information on SHAP features, Figure 4 shows that we do not differentiate between multiple features since we used only data that is available by the BraTS dataset. Therefore, as input features of our ML approach, we use age and a 3D MRI voxel given by the BraTS training dataset. The CNN used for predicting the survival rate has additional feature layers which are used to learn features retrieved from the input during learning. These features are however not extractable using SHAP.
Point 7: 340-343: Is there a general pattern that SHAPE features overlap with the extracted biological information? How well is this overlap? Could you give an overall statistic or measure it besides prediction accuracy or case-by-case eye-balling evaluation like in the figures given? Please provide the following info in the figures or figure captions: what is the ground truth for the survival and what was predicted.
We only looked at 29 randomly selected samples of the subset to test whether SHAP overlays can be used to manually get insights. Within these 29 samples we checked whether overlays make sense at all and got a feeling which parts relate to which feature importance value. We describe excerpts for two selected cases in Figure 6 and 7.
We added information regarding validation in line 308 and 297+.
Point 8: Quality of the figures are not good and the sizes of the same result figures are not consistent.
We adapted the sizes of the Figure to have the same size. The quality of the images (7-9) depends on the output of the trained network and SHAP, which restricts the resolution of the background image. Table 1 states the input sizes of the voxels for training the networks. These sizes limit the maximal resolution of the visualization output.
Point 9: 369-373: Again, evaluation of a model cannot be done by visualizing the learned features on training data. Authors either should change these claims or evaluate the model’s overall quality by validation and using standard metrics on validation data.
As discussed above we used standard metrics for validation; We “only” additionally use shapley values to support the evaluation and not as independent validation mechanism. We refined our descriptions.
Point 10: There are some easier to address concerns and here are my suggestions:
Ad. Abstract: There is an abbreviation for artificial intelligence (AI) but not for MRI, BraTS, and SHAP. Make sure to write long versions of the abbreviations where they are used for the first time, especially in the abstract.
We again checked abbreviations and corrected first usages to include the long versions.
Ad. Introduction: Line 54: reference for SHAP is missing here when it is first mentioned.
We added the reference at the first mention of SHAP.
Ad. Line 105, 108, 112, 113, 131, 160, 167: Sentences start with the reference number, and the numbers are used as a subject of the sentences. I think using passive sentence structure and adding the references at the end of the sentences will improve the writing of the manuscript.
The references and sentences were adapted accordingly so that no plain number starts the sentence.
Ad. Line 137 -> Line 62: Give long version of XAI earlier in the text when it is first used: move “explainable AI (XAI)” above line 62 from line 137.
We corrected redundant long versions and the long version of XAI is already mentioned before (line 43).
Ad. Line 142: Causability -> causability
The typo was fixed.
Ad. Line 145: What is DIN EN ISO 9241-11? Add a few words of definition to the text.
DIN EN ISO 9241-11 was defined further in the text as a norm that describes the ergonomics in human machine interaction.
Ad. 162: move “SHapley Additive exPlanations (SHAP)” where you first mention SHAP in abstract.
The abbreviation in the abstract was corrected.
Ad. 188: What is (r p y)? Add some descriptive text such as, vertical axis (yaw), transverse axis (pitch), longitudinal axis (roll).
We’ve added some descriptions on the variable names as suggested.
Ad. 208: “the CNN is trained on only one type…” which one?
Here we refer to the 2d full size model and corrected the unclear description in the manuscript.
Ad. 238 -> 239, no need for new paragraph. And some of these descriptions can also be mentioned in the intro where you introduce your paper’s main points.
Respective paragraph was removed.
239-246: I found this paragraph very difficult to follow. It could be explained step-by-step how the explainability was applied and I think this is the most important part of this article and should be told clearer.
The process of the application of DeepExplainer is described further in this section.
Ad. 250-251: it would be a good idea to include overall survival prediction in the figures (figure 4, also figure 6-9) since the prediction evaluation is done using overall survival.
The survival information (predicted and actual) were added to the figures.
Ad. 253-254: “As described in Figure 2, age is also included as a feature in the training, but the visualization of this feature is not done in SHAP”. I do not see any description mentioned above in figure 2. And please include here, the result for age feature and the reference figure for the visualization of the age feature.
We corrected the mistake and removed the whole sentence, as age features are actually visualized by SHAP.
Ad. 263: About the performance metrics, what is the algorithms accuracy performances per class, T0, T1, and T_3.
For a better understanding of the results, we’ve included the class accuracies for all three classes in a new table (Table 4).
Ad. Table 2: It is very difficult to understand if a prediction is good or not or compare with another prediction result unless the evaluation metrics are normalized. I suggest authors to report normalized root mean squared error (NMRSE) for the regression problem for example.
To compare the test results to the validation data, we’ve included only metrics provided by the MICCAI leaderboard. We did not choose the NMRSE, as this indicator shows to be not reliable for small sample sizes (see as example: Hanna et al. “Development and Application of a Simple Method for Evaluating Air Quality”). Again we clarified in the corresponding section(s).
Ad. Discussion: ---I think per class accuracies are more informative and classification results could be better per class
We’ve included class accuracies in the discussion as well.
Ad. 376-377: Is it possible to discover new useful information using SHAP features?
As mentioned about the SHAP framework, it provides additive feature attribution methods and visualization of these, such as importance, to support information discovery. Figure 4 illustrates such possibilities. Useful information may include insights into classification decisions and reproducibility.
Ad. Conclusion: 384-385: Do you have any examples/results that SHAPE values are helpful interpreting predictions on validation set? Can they be generalized? Like rule extraction?
The examples which we have are exemplarily shown for the test set, as the leaderboard has the ground truth data available to validate the interpretation. Validation data could be analyzed similarly, but assumptions on the interpretation cannot be validated.
Ad. 394-395: What are the three-class distribution in the training and test data? Did you use stratified sampling? If yes, please report this information and details in the text. If not, it could be the reason for overfitting and poor performance with validation data.
The class distributions on the training and test dataset were selected as analyzed by the dataset providers (see reference [73], Bakas et al. (2018) “Identifying the Best Machine Learning Algorithms for Brain Tumor Segmentation, Progression Assessment, and Overall Survival Prediction in the BRATS Challenge”). To quote these authors “These thresholds were derived after statistical consideration of the survival distributions across the complete dataset. Specifically, we chose these thresholds based on equal quantiles from the median OS (approximately 12.5 months) to avoid potential bias towards one of the survival groups (short- vs long- survivors) and while considering that discrimination of groups should be clinically meaningful.”
Ad. Some references are not up to date, there are arxiv papers from 2017 but they are published in the journals already. Like the most important reference for this paper which is the paper of SHAP (SHapley Additive exPlanations) [60]
Thank you for this comment. We corrected several former arxiv references and added updated publications.
Round 2
Reviewer 1 Report
Paper can be accepted
Author Response
We would like to thank you again for your time to help us refine our manuscript.
We further added some more improvements to the manuscript, highlighted in red.
Reviewer 2 Report
The authors have satisfactorily replied to my queries.
Author Response
We would like to thank you again for your time to help us refine our manuscript.
Just ot be informed, we further added some more improvements to the manuscript, highlighted in red.
Reviewer 3 Report
Authors used regression CNN to predict brain cancer patients’ survival from their brain tumor images and used SHAP values to provide visual interpretation of the model’s decisions during training. Overall, the manuscript has been improved after the revision. I would like to make some minor suggestions, which I believe could be used to improve the manuscript.
- What is t1, t1ce, t2 and flair in MRI, other than being different image modalities? I think it would be beneficial to mention shortly what they measure how they differ.
- Authors calculated accuracy measures separately for each of the aforementioned scan modalities. When visualizing the SHAP values, the examples in the manuscript are for FLAIR modality. Did authors look at other modalities’ SHAP values and how successful is the model’s interpretability comparing all the modalities t1, t1ce, t2?
- Authors response: “We only looked at 29 randomly selected samples of the subset to test whether SHAP overlays can be used to manually get insights. Within these 29 samples we checked whether overlays make sense at all and got a feeling which parts relate to which feature importance value. We describe excerpts for two selected cases in Figure 6 and 7.”
To make sense of the training network and provide a generalizable interpretation, other than looking at sample result by sample result, is it possible to look if there is an overall pattern that algorithm learns from training where ground truth available (hypothetically, for example, for patients who have low survival, model assigned larger weights to the border of the tumor in the image and for patients who have longer survival, model learned higher weights for the pixels lie inside the tumor etc.)?
- Authors perform CNN regression to predict survival, then results are evaluated using different metrics, (MSE as well as accuracy after the continuous predictions converted to classes). When the error is not normalized/comparable, it is difficult to make a conclusion on the quality of the prediction. I suggest authors to provide a scatter plot of actual versus predicted survival rate for survival prediction results.
Line 157: mdecial -> medical
Line 314 brats -> BraTS
Line 331 “… we refer to the MICCAI-BraTS leaderbordwhich lists the results for predicting the survival rate by multiple teams...” -> please add a reference here or a link if this is website
Author Response
We would like to thank you again for your time to help us refine our manuscript. All changes are highlighted in red.
- Ad. What is t1, t1ce, t2 and flair in MRI, other than being different image modalities? I think it would be beneficial to mention shortly what they measure how they differ.
Thank you for this comment, we added a short description to the introduction.
- Ad. Authors calculated accuracy measures separately for each of the aforementioned scan modalities. When visualizing the SHAP values, the examples in the manuscript are for FLAIR modality. Did authors look at other modalities’ SHAP values and how successful is the model’s interpretability comparing all the modalities t1, t1ce, t2?
We had a look at other image modalities too, However, flair worked best as example visualization modality regarding contrast and visibility. Furthermore, training with flair modalities showed best performance, therefore suggested further interpreting this type. We added a description and statement to the results and conclusion section.
- Ad. To make sense of the training network and provide a generalizable interpretation, other than looking at sample result by sample result, is it possible to look if there is an overall pattern that algorithm learns from training where ground truth available (hypothetically, for example, for patients who have low survival, model assigned larger weights to the border of the tumor in the image and for patients who have longer survival, model learned higher weights for the pixels lie inside the tumor etc.)?
In this work we focused on explainability methods as an addition to predictive AI on survival in glioma patients, while we emphasize interpretability as a qualitative task. Such qualitative experiments may suggest that there are more irrelevant features from pre-processing artifacts as well as contributing features such as tumor resection areas. However, deriving patterns is a quantitative research that needs way more experimentation and exceeds this work’s scope. Of course, this is one of many interesting future questions, such as quantitative expert testing, or one may use some of theVASARI (Visually AcceSAble Rembrandt Images) MRI feature set for comparison with SHAP overlays.
We further added a note regarding this comment to the discussion and conclusion section.
- Ad. Authors perform CNN regression to predict survival, then results are evaluated using different metrics, (MSE as well as accuracy after the continuous predictions converted to classes). When the error is not normalized/comparable, it is difficult to make a conclusion on the quality of the prediction. I suggest authors to provide a scatter plot of actual versus predicted survival rate for survival prediction results.
Thank you for your suggestion, we added scatterplots to the results section.
- Ad. Line 157: mdecial -> medical, Line 314 brats -> BraTS, Line 331 “… we refer to the MICCAI-BraTS leaderbordwhich lists the results for predicting the survival rate by multiple teams...” -> please add a reference here or a link if this is website
Thank you for your thorough review, we corrected the spelling mistakes and added the related information!